# Simple and Controllable Music Generation

Jade Copet♠◇    Felix Kreuk♠◇    Itai Gat    Tal Remez    David Kant
Gabriel Synnaeve ◇    Yossi Adi*◇    Alexandre Défossez ◇

♠: equal contributions, ◇: core team
Meta AI
{jadecopet, felixkreuk, adiyoss}@meta.com

## Abstract

We tackle the task of conditional music generation. We introduce MUSICGEN, a single Language Model (LM) that operates over several streams of compressed discrete music representation, i.e., tokens. Unlike prior work, MUSICGEN is comprised of a single-stage transformer LM together with efficient token interleaving patterns, which eliminates the need for cascading several models, e.g., hierarchically or up-sampling. Following this approach, we demonstrate how MUSICGEN can generate high-quality samples, both mono and stereo, while being conditioned on textual description or melodic features, allowing better controls over the generated output. We conduct extensive empirical evaluation, considering both automatic and human studies, showing the proposed approach is superior to the evaluated baselines on a standard text-to-music benchmark. Through ablation studies, we shed light over the importance of each of the components comprising MUSICGEN. Music samples, code, and models are available at github.com/facebookresearch/audiocraft.

## 1    Introduction

Text-to-music is the task of generating musical pieces given text descriptions, e.g., "90s rock song with a guitar riff". Generating music is a challenging task as it requires modeling long range sequences. Unlike speech, music requires the use of the full frequency spectrum [Müller, 2015]. That means sampling the signal at a higher rate, i.e., the standard sampling rates of music recordings are 44.1 kHz or 48 kHz vs. 16 kHz for speech. Moreover, music contains harmonies and melodies from different instruments, which create complex structures. Human listeners are highly sensitive to disharmony [Fedorenko et al., 2012, Norman-Haignere et al., 2019], hence generating music does not leave a lot of room for making melodic errors. Lastly, the ability to control the generation process in a diverse set of methods, e.g., key, instruments, melody, genre, etc. is essential for music creators.

Recent advances in self-supervised audio representation learning [Balestriero et al., 2023], sequential modeling [Touvron et al., 2023], and audio synthesis [Tan et al., 2021] provide the conditions to develop such models. To make audio modeling more tractable, recent studies proposed representing audio signals as multiple streams of discrete tokens representing the same signal [Défossez et al., 2022]. This allows both high-quality audio generation and effective audio modeling. However, this comes at the cost of jointly modeling several parallel dependent streams.

Kharitonov et al. [2022], Kreuk et al. [2022] proposed modeling multi-streams of speech tokens in parallel following a delay approach, i.e., introduce offsets between the different streams. Agostinelli et al. [2023] proposed representing musical segments using multiple sequences of discrete tokens at different granularity and model them using a hierarchy of autoregressive models. In parallel, Donahue et al. [2023] follows a similar approach but for the task of singing to accompaniment generation. Recently, Wang et al. [2023] proposed tackling this problem in two stages: (i) modeling the first

---

*Yossi Adi is Affiliated with both The Hebrew University of Jerusalem & MetaAI.

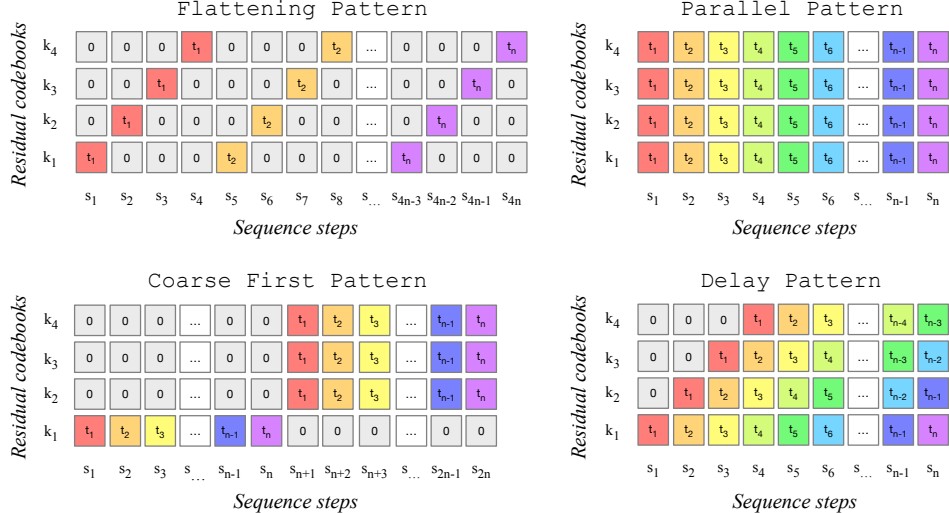

Figure 1: Codebook interleaving patterns presented in Section 2.2. Each time step $t_1, t_2, \ldots, t_n$ is composed of 4 quantized values (corresponding to $k_1, \ldots, k_4$). When doing autoregressive modelling, we can flatten or interleave them in various ways, resulting in a new sequence with 4 parallel streams and steps $s_1, s_2, \ldots, s_m$. The total number of sequence steps $S$ depends on the pattern and original number of steps $T$. 0 is a special token indicating empty positions in the pattern.

stream of tokens only; (ii) then, apply a post-network to jointly model the rest of the streams in a non-autoregressive manner.

In this work, we introduce MUSICGEN, a simple and controllable music generation model, which is able to generate high-quality music given textual description. We propose a general framework for modeling multiple parallel streams of acoustic tokens, which serves as a generalization of previous studies (see Figure 1). We show how this framework allows to extend generation to stereo audio at no extra computational cost. To improve controllability of the generated samples, we additionally introduce unsupervised melody conditioning, which allows the model to generate music that matches a given harmonic and melodic structure. We conduct an extensive evaluation of MUSICGEN and show the proposed method is superior to the evaluated baselines by a large margin, with a subjective rating of 84.8 out of 100 for MUSICGEN against 80.5 for the best baseline. We additionally provide an ablation study which sheds light on the importance of each of the components on the overall model performance. Lastly, human evaluation suggests that MUSICGEN yields high quality samples which are better melodically aligned with a given harmonic structure, while adhering to a textual description.

**Our contribution:** (i) We introduce a simple and efficient model to generate high quality music at 32 kHz. We show that MUSICGEN can generate consistent music with a single-stage language model through an efficient codebook interleaving strategy. (ii) We present a single model to perform both text and melody-conditioned generation and demonstrate that the generated audio is coherent with the provided melody and faithful to the text conditioning information. (iii) We provide extensive objective and human evaluations on the key design choices behind our method.

## 2 Method

MUSICGEN consists in an autoregressive transformer-based decoder [Vaswani et al., 2017], conditioned on a text or melody representation. The (language) model is over the quantized units from an EnCodec [Défossez et al., 2022] audio tokenizer, which provides high fidelity reconstruction from a low frame rate discrete representation. Compression models such as [Défossez et al., 2022, Zeghidour et al., 2021] employ Residual Vector Quantization (RVQ) which results in several parallel streams. Under this setting, each stream is comprised of discrete tokens originating from different learned codebooks. Prior work, proposed several modeling strategies to handle this issue [Kharitonov et al., 2022, Agostinelli et al., 2023, Wang et al., 2023]. In this work, we introduce a novel modeling framework, which generalizes to various codebook interleaving patterns, and we explore several

variants. Through patterns, we can leverage the internal structure of the quantized audio tokens. Finally, MUSICGEN supports conditional generation based on either text or melody.

## 2.1 Audio tokenization

We use EnCodec [Défossez et al., 2022], a convolutional auto-encoder with a latent space quantized using Residual Vector Quantization (RVQ) [Zeghidour et al., 2021], and an adversarial reconstruction loss. Given a reference audio random variable $X \in \mathbb{R}^{d \cdot f_s}$ with $d$ the audio duration and $f_s$ the sample rate, EnCodec encodes it into a continuous tensor with a frame rate $f_r \ll f_s$. This representation is then quantized into $Q \in \{1, \ldots, M\}^{d \cdot f_r \times K}$, with $K$ being the number of codebooks used in RVQ and $M$ being the codebook size. Notice, after quantization we are left with $K$ parallel discrete tokens sequences, each of length $T = d \cdot f_r$, representing the audio sample. In RVQ, each quantizer encodes the quantization error left by the previous quantizer, thus quantized values for different codebooks are in general not independent, and the first codebook is the most important one.

## 2.2 Codebook interleaving patterns (see Figure 1)

**Exact flattened autoregressive decomposition.** An autoregressive model requires a discrete random sequence $U \in \{1, \ldots, M\}^S$ with $S$ the sequence length. By convention, we will take $U_0 = 0$, a deterministic special token indicating the beginning of the sequence. We can then model the distribution

$$\forall t > 0, p_t (U_{t-1}, \ldots, U_0) \triangleq \mathbb{P} [U_t | U_{t-1}, \ldots, U_0]. \tag{1}$$

Let us build a second sequence of random variables $\tilde{U}$ using the auto-regressive density $p$, e.g. we define recursively $\tilde{U}_0 = 0$, and for all $t > 0$,

$$\forall t > 0, \mathbb{P} \left[ \tilde{U}_t | \tilde{U}_{t-1} \ldots, \tilde{U}_0 \right] = p_t \left( \tilde{U}_{t-1}, \ldots, \tilde{U}_0 \right). \tag{2}$$

Then, we immediately have that $U$ and $\tilde{U}$ follow the same distribution. This means that if we can fit a perfect estimate $\hat{p}$ of $p$ with a deep learning model, then we can fit exactly the distribution of $U$.

As stated before, the main issue with the representation $Q$ we obtained from the EnCodec model is that there are $K$ codebooks for each time step. One solution would be to flatten out $Q$, thus taking $S = d \cdot f_r \cdot K$, e.g. first predicting the first codebook of the first time step, then the second codebook of the first time step, etc. Then, using eq. (1) and eq. (2), we could theoretically fit an exact model of the distribution of $Q$. The downside however is the increased complexity, with part of the gain coming from the lowest sample rate $f_r$ being lost.

More than one possible flattening exists, and not all the $\hat{p}_t$ functions need to be estimated through a single model. For instance, MusicLM [Agostinelli et al., 2023] uses two models, one modeling the flattened first $K/2$ codebooks, and a second one the other $K/2$ flattened codebooks, conditioned on the decision of the first model. In that case, the number of autoregressive steps is still $df_r \cdot K$.

**Inexact autoregressive decomposition.** Another possibility is to consider an autoregressive decomposition, where some codebooks are predicted in *parallel*. For instance, let us define another sequence with $V_0 = 0$ and for all $t \in \{1, \ldots, T\}$, $k \in \{1, \ldots, K\}$, $V_{t,k} = Q_{t,k}$. When dropping the codebook index $k$, e.g. $V_t$, we mean the concatenation of all the codebooks at time $t$.

$$p_{t,k} (V_{t-1}, \ldots, V_0) \triangleq \mathbb{P} [V_{t,k} | V_{t-1}, \cdot, \ldots, V_0]. \tag{3}$$

Let's define again recursively $\tilde{V}_0 = 0$ and for all $t > 0$,

$$\forall t > 0, \forall k, \mathbb{P} \left[ \tilde{V}_{t,k} \right] = p_{t,k} \left( \tilde{V}_{t-1}, \ldots, \tilde{V}_0 \right). \tag{4}$$

Unlike in (2), we no longer have in the general case that $\tilde{V}$ follows the same distribution as $V$, even assuming we have access to the exact distribution $p_{t,k}$. In fact, we would only have a proper generative model if for all $t$, $(V_{t,k})_k$ are independent conditionally on $V_{t-1}, \ldots, V_0$. As $t$ increases, the errors will compound and the two distributions can grow further apart. Such a decomposition is *inexact*, but allows to keep the original frame rate which can considerably speed up training and inference, especially for long sequences.

**Arbitrary codebook interleaving patterns.** In order to experiment with various such decompositions, and measure exactly the impact of using an inexact decomposition, we introduce *codebook interleaving patterns*. Let us consider $\Omega = \{(t, k) : \{1, \ldots, d \cdot f_r\}, k \in \{1, \ldots, K\}\}$ be the set of all pairs of time steps and codebook indexes. A codebook pattern is a sequence $P = (P_0, P_1, P_2, \ldots, P_S)$, with $P_0 = \emptyset$, and for all $0 < s \le S$, $P_s \subset \Omega$, such that $P$ is partition of $\Omega$. We model $Q$ by predicting in parallel all the positions in $P_s$, conditionally on all the positions in $P_0, P_1, \ldots, P_{s-1}$. Pragmatically, we restrict ourselves to patterns where each codebook index appears at most once in any of the $P_s$.

We can now easily define a number of decompositions, for instance the "parallel" pattern given by

$$P_s = \{(s, k) : k \in \{1, \ldots, K\}\}. \tag{5}$$

It is also possible to introduce a "delay" between the codebooks, as in Kharitonov et al. [2022], e.g.,

$$P_s = \{(s - k + 1, k) : k \in \{1, \ldots, K\}, s - k \ge 0\}. \tag{6}$$

Through empirical evaluations, we show the benefits and drawbacks of various codebook patterns, shedding light on the importance of exact modeling of the parallel codebook sequences.

## 2.3 Model conditioning

**Text conditioning.** Given a textual description matching the input audio $X$, we compute a conditioning tensor $C \in \mathbb{R}^{T_C \times D}$ with $D$ being the inner dimension used in the autoregressive model. Generally, there are three main approaches for representing text for conditional audio generation. Kreuk et al. [2022] proposed using a pretrained text encoder, specifically T5 [Raffel et al., 2020]. Chung et al. [2022] show that using instruct-based language models provide superior performance. Lastly, Agostinelli et al. [2023], Liu et al. [2023], Huang et al. [2023a], Sheffer and Adi [2023] claimed that joint text-audio representation, such as CLAP [Wu* et al., 2023], provides better-quality generations. We experiment with all of the above, respectively: T5 encoder, FLAN-T5, and CLAP.

**Melody conditioning.** While text is the prominent approach in conditional generative models nowadays, a more natural approach for music is conditioning on a melodic structure from another audio track or even whistling or humming. Such an approach also allows for an iterative refinement of the model's output. To support that, we experiment with controlling the melodic structure via jointly conditioning on the input's chromagram and text description. In preliminary experiments, we observed that conditioning on the raw chromagram often led to reconstructing the original sample, resulting in overfitting. To reduce it, we introduce an information bottleneck by choosing the dominant time-frequency bin in each time step. While a similar capability was shown in Agostinelli et al. [2023], the authors used supervised proprietary data, which is tedious and costly to collect. In this work, we take an unsupervised approach, eliminating the requirement for supervised data.

## 2.4 Model architecture

**Codebook projection and positional embedding.** Given a codebook pattern, only some codebooks are present at each pattern step $P_s$. We retrieve from $Q$ the values corresponding to the indices in $P_s$. As noted in Section 2.2, each codebook is present at most once in $P_s$ or not at all. If it is present, we use a learned embedding table with $N$ entries and dimension $D$ to represent the associated value from $Q$. Otherwise, we use a special token indicating its absence. We sum the contribution from each codebook after this transformation. As $P_0 = \emptyset$, the first input is always the sum of all the special tokens. Finally, we sum a sinusoidal embedding to encode the current step $s$ [Vaswani et al., 2017].

**Transformer decoder.** The input is fed into a transformer with $L$ layers and a dimension $D$. Each layer consists of a causal self-attention block. We then use a cross-attention block that is fed with the conditioning signal $C$. When using melody conditioning, we instead provide the conditioning tensor $C$ as a prefix to the transformer input. The layer ends with a fully connected block consisting of a linear layer from $D$ to $4 \cdot D$ channels, a ReLU, and a linear layer back to $D$ channels. The attention and fully connected blocks are wrapped with a residual skip connection. Layer normalization [Ba et al., 2016] is applied to each block before being summed with the residual skip connection ("pre-norm").

**Logits prediction.** The output from the transformer decoder at pattern step $P_s$ is transformed into logits prediction for the values of $Q$ taken at the indices given by $P_{s+1}$. Each codebook is present at most once in $P_{s+1}$. If a codebook is present, the logits prediction is obtained by applying a codebook specific linear layer from $D$ channels to $N$.

# 3 Experimental setup

## 3.1 Models and hyperparameters

**Audio tokenization model.** We use a non-causal five layers EnCodec model for 32 kHz monophonic audio with a stride of 640, resulting in a frame rate of 50 Hz, and an initial hidden size of 64, doubling at each of the model's five layers. The embeddings are quantized with a RVQ with four quantizers, each with a codebook size of 2048. We follow Défossez et al. [2022] to train the model on one-second audio segments cropped at random in the audio sequence.

**Transformer model.** We train autoregressive transformer models at different sizes: 300M, 1.5B, 3.3B parameters. We use a memory efficient Flash attention [Dao et al., 2022] from the xFormers package [Lefaudeux et al., 2022] to improve both speed and memory usage with long sequences. We study the impact of the size of the model in Section 4. We use the 300M-parameter model for all of our ablations. We train on 30-second audio crops sampled at random from the full track. We train the models for 1M steps with the AdamW optimizer [Loshchilov and Hutter, 2017], a batch size of 192 examples, $\beta_1 = 0.9$, $\beta_2 = 0.95$, a decoupled weight decay of 0.1 and gradient clipping of 1.0. We further rely on D-Adaptation based automatic step-sizes [Defazio and Mishchenko, 2023] for the 300M model as it improves model convergence but showed no gain for the bigger models. We use a cosine learning rate schedule with a warmup of 4000 steps. Additionally, we use an exponential moving average with a decay of 0.99. We train the 300M, 1.5B and 3.3B parameter models, using respectively 32, 64 and 96 GPUs, with mixed precision. More specifically, we use float16 as bfloat16 was leading to instabilities in our setup. Finally, for sampling, we employ top-k sampling [Fan et al., 2018] with keeping the top 250 tokens and a temperature of 1.0.

**Text preprocessing.** Kreuk et al. [2022] proposed a text normalization scheme, in which stop words are omitted and the remaining text is lemmatized. We denote this method by text-normalization. When considering musical datasets, additional annotations tags such as musical key, tempo, type of instruments, etc. are often available. We also experiment with concatenating such annotations to the text description. We denote this approach by condition-merging. Finally, we explored using word dropout as another text augmentation strategy. For the final models, we used condition-merging with a probability of 0.25. Upon merging, we apply a text description dropout with a probability of 0.5. We use a word dropout with a probability of 0.3 on the resulting text. A full comparison of the different text preprocessing strategies can be found in Appendix A.2.

**Codebook patterns and conditioning.** We use the "delay" interleaving pattern from Section 2.2, This translates 30 seconds of audio into 1500 autoregressive steps. For text conditioning, we use the T5 [Raffel et al., 2020] text encoder, optionally with the addition of the melody conditioning presented in Section 2.3. We also experiment with FLAN-T5 [Chung et al., 2022], and CLAP [Wu* et al., 2023] and compare the performance of MUSICGEN using each of these text encoders in the Appendix A.2. For melody conditioning, we compute the chromagrams with a window size of $2^{14}$ and a hop size of $2^{12}$. Using a large window prevents the model from recovering fine temporal details. We further quantize the chromagram by taking the argmax at each time step. We follow a similar approach to Kreuk et al. [2022] and implement classifier-free guidance when sampling from the model's logits. Specifically, during training we drop the condition with a probability of 0.2 and during inference we use a guidance scale of 3.0.

## 3.2 Datasets

**Training datasets.** We use 20K hours of licensed music to train MUSICGEN. Specifically, we rely on an internal dataset of 10K high-quality music tracks, and on the ShutterStock and Pond5 music data collections[2] with respectively 25K and 365K instrument-only music tracks. All datasets consist of full-length music sampled at 32 kHz with metadata composed of a textual description and information such as the genre, BPM, and tags. We downmix the audio to mono unless stated otherwise.

**Evaluation datasets.** For the main results and comparison with prior work, we evaluate the proposed method on the MusicCaps benchmark [Agostinelli et al., 2023]. MusicCaps is composed of 5.5K samples (ten-second long) prepared by expert musicians and a 1K subset balanced across genres. We report objective metrics on the unbalanced set, while we sample examples from the genre-balanced

---

[2]www.shutterstock.com/music and www.pond5.com

Table 1: Text-to-Music generation. We compare objective and subjective metrics for MUSICGEN against a number of baselines. We report both mean and CI95 scores. The Mousai model is retrained on the same dataset, while for MusicLM we use the public API for human studies. We report the original FAD on MusicCaps for Noise2Music and MusicLM. "MUSICGEN w. random melody" refers to MUSICGEN trained with chromagram and text. At evaluation time, we sample the chromagrams at random from a held-out set.

| | MUSICCAPS Test Set | | | | |
|---|---|---|---|---|---|
| MODEL | $\text{FAD}_{vgg} \downarrow$ | KL $\downarrow$ | $\text{CLAP}_{scr} \uparrow$ | OVL. $\uparrow$ | REL. $\uparrow$ |
| Riffusion | 14.8 | 2.06 | 0.19 | $79.31_{\pm 1.37}$ | $74.20_{\pm 2.17}$ |
| Mousai | 7.5 | 1.59 | 0.23 | $76.11_{\pm 1.56}$ | $77.35_{\pm 1.72}$ |
| MusicLM | 4.0 | - | - | $80.51_{\pm 1.07}$ | $82.35_{\pm 1.36}$ |
| Noise2Music | **2.1** | - | - | - | - |
| MUSICGEN w.o melody (300M) | 3.1 | 1.28 | 0.31 | $78.43_{\pm 1.30}$ | $81.11_{\pm 1.31}$ |
| MUSICGEN w.o melody (1.5B) | 3.4 | 1.23 | **0.32** | $80.74_{\pm 1.17}$ | **$83.70_{\pm 1.21}$** |
| MUSICGEN w.o melody (3.3B) | 3.8 | **1.22** | 0.31 | **$84.81_{\pm 0.95}$** | $82.47_{\pm 1.25}$ |
| MUSICGEN w. random melody (1.5B) | 5.0 | 1.31 | 0.28 | $81.30_{\pm 1.29}$ | $81.98_{\pm 1.79}$ |

set for qualitative evaluations. For melody evaluation and the ablation studies, we use samples from an in-domain held out evaluation set of $528$ music tracks, with no artist overlap with the training set.

## 3.3 Evaluation

**Baselines.** We compare MUSICGEN to two baselines for text-to-music generation: Riffusion [Forsgren and Martiros] and Mousai [Schneider et al., 2023]. We use the open source Riffusion model to run inference [3]. For Mousai, we train a model using our dataset for a fair comparison, using the open source implementation provided by the authors[4]. Additionally, when possible, we compare to MusicLM [Agostinelli et al., 2023] and Noise2Music [Huang et al., 2023b].

**Evaluation metrics.** We evaluate the proposed method using objective and subjective metrics. For the objective methods, we use three metrics: the Fréchet Audio Distance (FAD), the Kullback-Leiber Divergence (KL) and the CLAP score (CLAP). We report the FAD [Kilgour et al., 2018] using the official implementation in Tensorflow with the VGGish model [5]. A low FAD score indicates the generated audio is plausible. Following Kreuk et al. [2022], we use a state-of-the-art audio classifier trained for classification on AudioSet [Koutini et al., 2021] to compute the KL-divergence over the probabilities of the labels between the original and the generated music. The generated music is expected to share similar concepts with the reference music when the KL is low. Last, the CLAP score [Wu* et al., 2023, Huang et al., 2023a] is computed between the track description and the generated audio to quantify audio-text alignment, using the official pretrained CLAP model [6].

For the human studies, we follow the same setup as in Kreuk et al. [2022]. We ask human raters to evaluate two aspects of the audio samples (i) overall quality (OVL), and (ii) relevance to the text input (REL). For the overall quality test, raters were asked to rate the perceptual quality of the provided samples in a range of 1 to 100. For the text relevance test, raters were asked to rate the match between audio and text on a scale of 1 to 100. Raters were recruited using the Amazon Mechanical Turk platform. We evaluate randomly sampled files, where each sample was evaluated by at least 5 raters. We use the CrowdMOS package[7] to filter noisy annotations and outliers. We remove annotators who did not listen to the full recordings, annotators who rate the reference recordings less than $85$, and the rest of the recommended recipes from CrowdMOS [Ribeiro et al., 2011]. For fairness, all samples are normalized at $-14$dB LUFS [ITU-R, 2017].

---

[3]Using riffusion-model-v1 from github.com/riffusion/riffusion-app (on May 10, 2023)

[4]Implementation from github.com/archinetai/audio-diffusion-pytorch (March 2023)

[5]github.com/google-research/google-research/tree/master/frechet_audio_distance

[6]https://github.com/LAION-AI/CLAP

[7]http://www.crowdmos.org/download/

Table 2: We report cosine similarity between reference and generated melody (SIM.) and subjective metrics including alignment with the melody (MEL.). All results are reported for MUSICGEN 1.5B.

| | | | In Domain Test Set | | |
|---|---|---|---|---|---|
| TRAIN CONDITION | TEST CONDITION | SIM. ↑ | MEL. ↑ | OVL. ↑ | REL. ↑ |
| Text | Text | 0.10 | $64.44_{\pm0.83}$ | $82.18_{\pm1.21}$ | $81.54_{\pm1.22}$ |
| Text+Chroma | Text | 0.10 | $61.89_{\pm0.96}$ | $81.65_{\pm1.13}$ | $\mathbf{82.50}_{\pm0.98}$ |
| Text+Chroma | Text+Chroma | **0.66** | $\mathbf{72.87}_{\pm0.93}$ | $\mathbf{83.94}_{\pm1.99}$ | $80.28_{\pm1.06}$ |

# 4 Results

We start by presenting results of the proposed method on the task of text-to-music generation and compare MUSICGEN to prior work in the field. Next, we evaluate the ability of the proposed method to generate music conditioned on melodic features. We further show how to simply extend our codebook patterns for stereo audio generation. We conclude with an ablation study. Music samples, code, and models are available at github.com/facebookresearch/audiocraft.

## 4.1 Comparison with the baselines

Table 1 presents the comparison of the proposed method against Mousai, Riffusion, MusicLM, and Noise2Music. As there is no official implementation of Noise2Music, nor pre-trained model, we report only FAD on MusicCaps, which is reported in the Noise2Music manuscript. Similarly, MusicLM implementation is not public. We use the MusicLM public demo[8] for our subjective tests while reporting the FAD as reported by the authors. While the original MusicLM model is trained on data with vocals, the model behind the API is instrument-only. For the human study, we restrict ourselves to 40 instrument-only samples from MusicCaps. To prevent leakage in MUSICGEN trained with chromagram, we sample chromagrams at random from a held-out set during test time.

Results suggest that MUSICGEN performs better than the evaluated baselines as evaluated by human listeners, both in terms of audio quality and adherence to the provided text description. Noise2Music performs the best in terms of FAD on MusicCaps, followed by MUSICGEN trained with text conditioning. Interestingly, adding a melody conditioning degrades the objective metrics, however, it does not significantly affect human ratings, while still being superior to the evaluated baselines.

We notice that for the worst rated model, the FAD is correlated with the overall subjective rating, but it is not the case for the best rated models. We noticed that a large number of samples in MusicCaps [Agostinelli et al., 2023] contains a description stating that the recording is "noisy". It is possible that due to those noisy samples, improvements in the quality of the generated audio might deteriorate the FAD on MusicCaps once a certain quality threshold is achieved.

## 4.2 Melody evaluation

We evaluate MUSICGEN, conditioned jointly on textual and melodic representations, using objective and subjective metric on the held out evaluation set. For the objective evaluation, we introduce a new metric: chroma cosine-similarity, which measures the average cosine-similarity between frames corresponding to the same time steps, taken from the quantized chroma of the reference and the generated samples. We evaluate using 1000 randomly sampled files from a held-out set. To better evaluate the relation between the conditioned melody to the generated music, we introduce another human study. To that end, we present human raters with a reference musical piece, followed by a set of generated pieces. For each generated sample, the listeners are asked to rate how well the melody of the generated piece matches that of the reference on a scale of 1 to 100. We use 40 samples of 10 seconds at random from the held-out set. Results are reported in Table 2. Results suggest that MUSICGEN trained with chromagram conditioning successfully generates music that follows a given melody. Thus, allowing for better control over the generated output. Interestingly, MUSICGEN is robust to dropping the chroma at inference time with both OVL. and REL. staying roughly the same.

---

[8]https://blog.google/technology/ai/musiclm-google-ai-test-kitchen/

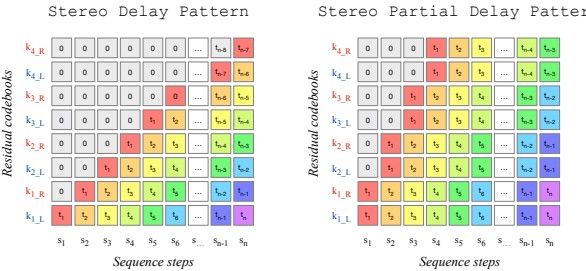
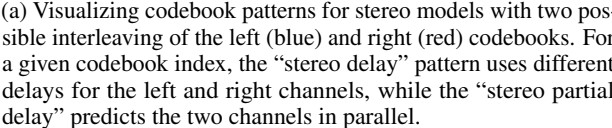

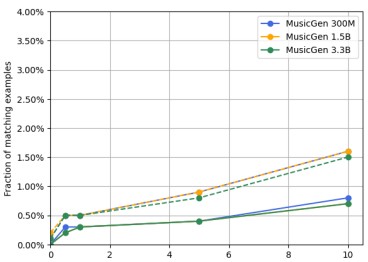

(a) Visualizing codebook patterns for stereo models with two possible interleaving of the left (blue) and right (red) codebooks. For a given codebook index, the "stereo delay" pattern uses different delays for the left and right channels, while the "stereo partial delay" predicts the two channels in parallel.

(b) Memorization results for the first codebook tokens on 5-second audio generations, considering exact (solid line) and 80% partial (dashed line) matches when prompted with extracts of varying duration from the train set.

Figure 2: Stereo codebooks (left) and memorization results (right)

Table 3: Stereophonic Text-to-Music generation. EnCodec processes separately the left and right channels, giving us 8 codebooks instead of 4. We experiment with two codebook patterns, depicted in Figure 2a. We also measure one of the stereo model after being downmixed to mono. We use a 1.5B MUSICGEN model conditioned only on text.

| Cb. Pattern | Stereo? | MUSICCAPS Test Set | |
|---|---|---|---|
| | | Ovl. ↑ | Rel. ↑ |
| *Mono Delay* | ✗ | $84.95_{\pm 1.60}$ | $\mathbf{80.61}_{\pm 1.22}$ |
| *Stereo Partial Delay* | ✗* | $84.49_{\pm 1.80}$ | $79.39_{\pm 1.16}$ |
| *Stereo Partial Delay* | ✓ | $\mathbf{86.73}_{\pm 1.06}$ | $80.41_{\pm 1.15}$ |
| *Stereo Delay* | ✓ | $85.51_{\pm 1.21}$ | $78.32_{\pm 1.21}$ |

*: downmixed to mono after generation.

### 4.3 Fine-tuning for stereophonic generation

We experiment with for extending generation to stereophonic data. We use the same EnCodec tokenizer, which we applied independently to the left and right channels, providing $2 \cdot K = 8$ codebooks per frame. Starting from a pre-trained monophonic MUSICGEN model, we fine tune it for 200K batches the same dataset with stereo audio. We reuse the "delay" pattern, with two possible variations: (i) "stereo delay" introduces a delay between the left and right channels for the same codebook level, while (ii) "stereo partial delay" apply the same delay to the left and right channel codebooks for a given level, as depicted in Figure 2a. Note that using this simple strategy, we can generate stereo audio *at no extra computational cost*. We provide in Table 3 the subjective evaluations for those models. We notice that when downmixing the stereo output to mono, we are almost equivalent in perceived quality to a mono model. Stereo audio was overall rated higher than the mono counterpart, and the "stereo partial delay" benefits from a small boost both in overall quality and text relevance compared with the "stereo delay" pattern.

### 4.4 Ablation

This section provides an ablation study for the different codebook patterns, together with results for model scales and a memorization study. Additionally, we present results for different text augmentation strategies, text encoders, and audio tokenization models in Appendix A.2. All ablations are performed using 1K samples of 30 seconds, randomly sampled from the held-out evaluation set.

**The effect of the codebook interleaving patterns.** We evaluate various codebook patterns using the framework from Section 2.2, with $K = 4$, given by the audio tokenization model. Table 1 reports results with the "delay" pattern. The "partial delay" consists in delaying by the same amount the codebooks 2, 3, and 4. The "parallel" pattern predicts all the codebooks from the same time step in parallel. The "coarse first" pattern first predicts codebook 1 for all steps, then predicts in parallel codebooks 2, 3, and 4. Thus, this pattern has twice the steps compared to other patterns. "Partial

Table 4: Codebook patterns. We compare different codebook interleaving patterns on 30-seconds, audio sequences. The "flattening" pattern achieves the best scores. The "delay" and "partial flattening" patterns achieve similar scores, while "parallel" obtains worse scores.

| CONFIGURATION | Nb. steps | In Domain Test Set | | | | |
| --- | --- | --- | --- | --- | --- | --- |
| | | $FAD_{vgg} \downarrow$ | $KL \downarrow$ | $CLAP_{scr} \uparrow$ | OVL. $\uparrow$ | REL. $\uparrow$ |
| Delay | 1500 | 0.96 | 0.52 | 0.35 | **79.69**$_{\pm1.46}$ | 79.67$_{\pm1.41}$ |
| Partial delay | 1500 | 1.51 | 0.54 | 0.32 | 79.13$_{\pm1.56}$ | 79.67$_{\pm1.46}$ |
| Parallel | 1500 | 2.58 | 0.62 | 0.27 | 72.21$_{\pm2.49}$ | 80.30$_{\pm1.43}$ |
| Partial flattening | 3000 | 1.32 | 0.54 | 0.34 | 78.56$_{\pm1.86}$ | 79.18$_{\pm1.49}$ |
| Coarse first | 3000 | 1.98 | 0.56 | 0.30 | 74.42$_{\pm2.28}$ | 76.55$_{\pm1.67}$ |
| Flattening | 6000 | **0.86** | **0.51** | **0.37** | **79.71**$_{\pm1.58}$ | **82.03**$_{\pm1.1}$ |

Table 5: Model scale. We compare 3 scales for our method, and evaluate it on an internal test set to limit the impact of the out of domain prediction issues we observed with MusicCaps. In terms of objective metrics we observe a continuous improvement of the metrics, although subjective quality stop improving at 1.5B parameters. A 3.3B model however seems to better fit the text prompt.

| Dim. | Heads | Depth | # Param. | In Domain Test Set | | | | | |
| --- | --- | --- | --- | --- | --- | --- | --- | --- | --- |
| | | | | PPL $\downarrow$ | $FAD_{vgg} \downarrow$ | $KL \downarrow$ | $CLAP_{scr} \uparrow$ | OVL. $\uparrow$ | REL. $\uparrow$ |
| 1024 | 16 | 24 | 300M | 56.1 | 0.96 | 0.52 | 0.35 | 78.3$_{\pm1.4}$ | 82.5$_{\pm1.6}$ |
| 1536 | 24 | 48 | 1.5B | 48.4 | 0.86 | **0.50** | 0.35 | **81.9**$_{\pm1.4}$ | 82.9$_{\pm1.5}$ |
| 2048 | 32 | 48 | 3.3B | **46.1** | **0.82** | **0.50** | **0.36** | 79.2$_{\pm1.3}$ | **83.5**$_{\pm1.3}$ |

flattening" is similar, but instead of sampling first codebook 1 for all steps, it interleaves them with the parallel sampling of codebooks 2, 3, and 4. Finally, the "flattening" pattern consists in flattening all the codebooks, similar to MusicLM [Agostinelli et al., 2023]. We report objective and subjective evaluations in Table 4. While flattening improves generation, it comes at a high computational cost and similar performance can be reached for a fraction of this cost using a simple delay approach.

**The effect of model size.** In Table 5 we report results for different model sizes, namely 300M, 1.5B, and 3.3B parameter models. As expected, scaling the model size results in better scores, however this comes at the expense of longer training and inference time. Regarding subjective evaluations, the overall quality is optimal at 1.5B, but a larger model can better understand the text prompt.

**Memorization experiment.** Following [Agostinelli et al., 2023], we analyze the memorization abilities of MUSICGEN. We only consider the first stream of codebooks from MUSICGEN as it contains the coarser-grain information. We randomly select $N = 20,000$ examples from our training set and for each one we feed the model with a prompt of EnCodec codebooks corresponding to the original audio and the conditioning information. We generate a continuation of 250 audio tokens (5 second-long audio) using greedy decoding. We report exact matches as the fraction of examples for which the generated audio tokens and source audio tokens are identical over the whole sequence. In addition, we report partial matches as the fraction of the training examples for which the generated and source sequences have at least $80\%$ of the audio tokens matching. We present the memorization results for the different model sizes when varying the length of the audio prompt in Figure 2b.

## 5   Related work

**Audio representation.** In recent years, the prominent approach is to represent the music signals in a compressed representation, discrete or continuous, and apply a generative model on top of it. Lakhotia et al. [2021] proposed quantizing speech representation using k-means to construct speech language models. Recently, Défossez et al. [2022], Zeghidour et al. [2021] proposed to apply a VQ-VAE directly on the raw waveform using residual vector quantization. Later, several studies used such representation for text-to-audio generation. Next, we discuss the recent research in audio generation.

**Music generation.** Music generation has been long studied under various setups. Dong et al. [2018] proposed a GAN-based approach for symbolic music generation. Bassan et al. [2022] proposed an unsupervised segmentation for symbolic music which can be later used for generation. Ycart et al.

[2017] proposed modeling polyphonic music modeling using recurrent neural networks. Ji et al. [2020] conducted a comprehensive survey therein for deep learning methods for music generation.

Dhariwal et al. [2020] proposed representing music samples in multiple streams of discrete representations using a hierarchical VQ-VAE. Next, two sparse transformers applied over the sequences to generate music. Gan et al. [2020] proposed generating music for a given video, while predicting its midi notes. Recently, Agostinelli et al. [2023] proposed representing music using multiple streams of "semantic tokens" and "acoustic tokens". Then, they applied a cascade of transformer decoders conditioned on a textual-music joint representation [Huang et al., 2022]. Donahue et al. [2023] followed a similar modeling approach, but for the task of singing-to-accompaniment generation.

An alternative approach is using diffusion models. Schneider et al. [2023], Huang et al. [2023b], Maina [2023], Forsgren and Martiros proposed using a latent diffusion model for the task of text-to-music. Schneider et al. [2023] proposed using diffusion models for both audio encoder-decoder and latent generation. Huang et al. [2023b] proposed a cascade of diffusion model to generate audio and gradually increase its sampling rate. Forsgren and Martiros proposed fine-tuning Stable Diffusion Rombach et al. [2022] using spectrograms to generate five-second segments, then, using image-to-image mapping and latent interpolation to generate long sequences.

**Audio generation.** Several studies were proposed for text-to-audio (environmental sounds) generation. Yang et al. [2022] proposed representing audio spectrograms using a VQ-VAE, then applying a discrete diffusion model conditioned on textual CLIP embeddings for the generation part [Radford et al., 2021]. Kreuk et al. [2022] proposed applying a transformer language model over discrete audio representation, obtained by quantizing directly time-domain signals using EnCodec [Défossez et al., 2022]. Sheffer and Adi [2023] followed a similar approach to Kreuk et al. [2022] for image-to-audio generation. Huang et al. [2023a], Liu et al. [2023] proposed using latent diffusion models for the task of text-to-audio, while extending it to various other tasks such as inpainting, image-to-audio, etc.

## 6    Discussion

We introduced MUSICGEN, a state-of-the-art single stage controllable music generation model that can be conditioned on text and melody. We demonstrated that simple codebook interleaving strategies can be used to achieve high quality generation, even in stereo, while reducing the number of autoregressive time steps compared to the flattening approach. We provided a comprehensive study of the impact of model sizes, conditioning methods, and text pre-processing techniques. We also introduced a simple chromagram-based conditioning for controlling the melody of the generated audio.

**Limitations** Our simple generation method does not allow us to have fine-grained control over adherence of the generation to the conditioning, we rely mostly on CF guidance. Also, while it is relatively straightforward to do data augmentation for text conditioning, conditioning on audio warrants further research on data augmentation, types and amount of guidance.

**Broader impact.** Large scale generative models present ethical challenges. We first ensured that all the data we trained on was covered by legal agreements with the right holders, in particular through an agreement with ShutterStock. A second aspect is the potential lack of diversity in the dataset we used, which contains a larger proportion of western-style music. However, we believe the simplification we operate in this work, e.g., using a single stage language model and a reduced number of auto-regressive steps, can help broaden the applications to new datasets. Generative models can represent an unfair competition for artists, which is an open problem. Open research can ensure that all actors have equal access to these models. Through the development of more advanced controls, such as the melody conditioning we introduced, we hope that such models can become useful both to music amateurs and professionals.

## Acknowledgements.

The authors would like to thank Mary Williamson, Rashel Moritz and Joelle Pineau for supporting this project, thank Justin Luk, Prash Jain, Sidd Srinivasan, Rod Duenes, and Philip Woods for the dataset, and thank the xformers team: Daniel Haziza, Francisco Massa, and Michael Ramamonjisoa for the technical discussions.

**Authors note.** This paper was submitted in the wake of the tragic terror attack perpetrated by Hamas on October 7, 2023, which has left the Israeli nation profoundly devastated. The assault greatly

impacted us at a personal level, thereby significantly impacting the course of this research work. This paper was finalized while we grieve and mourn our friends and family, under great stress, with scientific considerations being the last thing on our minds. It may contain subtle errors.

In memory of the countless lives shattered by Hamas actions.

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

# A  Appendix

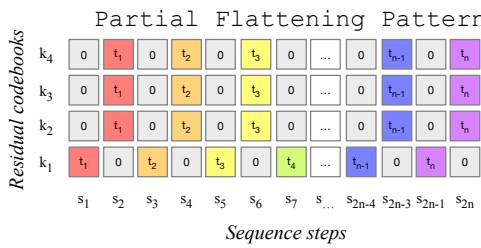 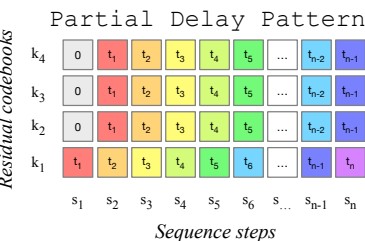

Figure A.1: Visualizing partial flattening and partial delays codebook patterns applied on a sequence with 4 parallel streams of quantized values (corresponding to $k_1, \ldots, k_4$) and $N$ time steps ($t_1, \ldots, k_n$). "Partial flattening" separates the first codebook in dedicated steps and interleaves them with the parallel sampling of codebooks 2, 3, and 4, leading the number of interleaved sequences steps $M$ to be twice the number of original steps $N$. The "partial delay" pattern consists in delaying by the same amount the codebooks 2, 3, and 4, in our case we use a delay of 1. The total number of steps of the interleave sequences is $N$ (excluding the last step for simplicity).

## A.1  Experimental details

**Codebook interleaving patterns.** Figure A.1 provides a visual description of the additional codebook patterns introduced for the ablation in Section 4, namely "partial flattening" and "partial delay" patterns. The intuition behind such patterns is driven by the fact that the first codebook from RVQ is the most important one and predicting the rest of the codebooks in parallel would allow to limit the introduced flattening or delay while maintaining good modeling performance.

**Melody conditioning.** In this work, we provide an unsupervised approach for melody conditioning through conditioning on the chromagram representation. As shown in Figure A.2, chromagram-based conditioning successfully preserves the melodic structure when generating novel music samples. In preliminary experiments, we noticed that the chromagram tends to be dominated by the lower frequency instruments, mainly by the drums and bass. To mitigate that, we used Demucs [Défossez et al., 2019] to first decompose the reference track into four components: drums, bass, vocals, and other. Next, we omit the drums and bass to recover the melodic structure of the residual waveform. Finally, we extract the quantized chromagram to create the conditioning that is later fed to the model.

**Distribution of genres.** We provide in Figure A.3 a histogram of the top 50 musical genres present in the dataset. We notice a clear dominance of the Dance/EDM genre, which in our experience is also one of the genres that is best generated by MUSICGEN. While we tried to explore a number of resampling scheme to boost the importance of other genres, we observed that oversampling less represented genres would often lead to a worse model overall.

## A.2  Additional experimental results

We provide further ablation studies on the core components of MUSICGEN, namely the text encoder used for text conditioning described in Section 2.3, text augmentation strategies presented in Section 3.1, and the used audio tokenization model. We report results on the MusicCaps dataset to better understand out-of-domain generalization abilities of the different approaches. Finally, we share additional experimental results on optimization methods.

**The effect of text encoder.** We investigate the impact of the text encoder, comparing three different encoders: T5 [Raffel et al., 2020] [9], Flan-T5 [Chung et al., 2022][10] and CLAP [Wu* et al., 2023] [11] with a quantization bottleneck. For the CLAP-based encoder, similarly to Agostinelli et al. [2023] we rely on the music embeddings during training and the text embeddings at inference time and

---

[9] https://huggingface.co/t5-base
[10] https://huggingface.co/google/flan-t5-base
[11] https://huggingface.co/lukewys/laion_clap/blob/main/music_audioset_epoch_15_esc_90.14.pt

Table A.1: Text encoder results. We report results for T5, Flan-T5, and CLAP as text encoders. We observe similar results for T5 and Flan-T5 on all the objective metrics. Note that T5 is the text encoder used for the main MUSICGEN models. CLAP encoder performs consistently worse on all the metrics but CLAP score. All comparisons are done with a 300M MUSICGEN model using text conditioning only.

| | MUSICCAPS Test Set | | | | |
| MODEL | $FAD_{vgg}\downarrow$ | KL $\downarrow$ | $CLAP_{scr}\uparrow$ | OVL. $\uparrow$ | REL. $\uparrow$ |
| --- | --- | --- | --- | --- | --- |
| T5 | **3.12** | **1.29** | 0.31 | $85.04_{\pm1.23}$ | $\textbf{87.33}_{\pm\textbf{1.9}}$ |
| Flan-T5 | 3.36 | 1.30 | 0.32 | $\textbf{85.54}_{\pm\textbf{1.01}}$ | $85.00_{\pm1.63}$ |
| CLAP | 4.16 | 1.36 | 0.35 | $82.13_{\pm1.29}$ | $83.56_{\pm1.54}$ |
| CLAP (no normalization) | 4.14 | 1.38 | 0.35 | $84.87_{\pm1.25}$ | $85.06_{\pm1.72}$ |
| CLAP (no quantization) | 5.07 | 1.37 | **0.37** | $84.13_{\pm1.02}$ | $84.67_{\pm1.42}$ |

Table A.2: Text augmentations strategies results. We report objective metrics using only the original text description (no augmentation) and for different text augmentation strategies: using condition merging to augment the text description with metadata, using text-normalization (text-norm.) and applying word dropout on the resulting text. We use 300M MUSICGEN models trained for 500K steps. Condition merging improves the result over training only over the original text description. Other augmentations perform worst on all metrics. We use the Condition Merging with Word dropout, showing the best text relevance, in our main models.

| | MUSICCAPS Test Set | | | | |
| CONFIGURATION | $FAD_{vgg}\downarrow$ | KL $\downarrow$ | $CLAP_{scr}$ | OVL. $\uparrow$ | REL. $\uparrow$ |
| --- | --- | --- | --- | --- | --- |
| No augmentation | 3.68 | 1.28 | **0.31** | $\textbf{83.40}_{\pm\textbf{1.44}}$ | $81.16_{\pm1.29}$ |
| Condition Merging (CM) | **3.28** | **1.26** | **0.31** | $82.60_{\pm1.41}$ | $84.45_{\pm1.16}$ |
| CM + Text-norm. (TN) | 3.78 | 1.30 | 0.29 | $80.57_{\pm2.14}$ | $82.40_{\pm1.09}$ |
| CM+ Word dropout (WD) | 3.31 | 1.31 | 0.30 | $82.52_{\pm1.55}$ | $\textbf{85.27}_{\pm\textbf{0.97}}$ |
| CM + TN + WD | 3.41 | 1.39 | 0.30 | $81.18_{\pm1.91}$ | $84.32_{\pm1.59}$ |

we train a RVQ layer on top of the extracted embeddings. More specifically, we first normalize the embeddings and use RVQ with 12 quantizers, each with a codebook size of 1024. Quantizing the CLAP embeddings leads to a homogeneous representation with the discrete tokens further reducing the gap between the audio encoding used at train time and text encoding at test time. We report results for the different text encoders in Table A.1. Both T5 and Flan-T5 perform similarly in terms of objective metrics, the overall quality being slighly better for T5. The CLAP-based model however suffers worse objective and subjective metrics, with the exception of the CLAP score which rely on the same underlying audio and text encoder.

**The effect of text augmentations.** We examine the impact of text augmentation strategies for the proposed method. In particular, we study the use of condition merging (i.e. concatenating additional metadata to the text description), text normalization (text-norm.) and word dropout. We report objective metrics for the different augmentation strategies in Table A.2. We observe a gain in FAD and KL when leveraging the additional metadata with condition merging. However, neither text normalization or word dropout improves the results in terms of objective metrics.

**The effect of the audio tokenizer.** We experiment with replacing EnCodec with Descript Audio Codec (DAC) [Kumar et al., 2023][12], a similar audio compression models, that uses a different training set, a similar adversarial loss enhanced with multiband discriminators, but performs quantization in a lower dimension space to improve codebook usage. DAC compresses audio at 44.1 kHz with 9 codebooks, and a framerate of 86 Hz. We trained a small (300M parameters) and a medium (1.5B parameters) MUSICGEN model using both DAC and EnCodec as an audio tokenizer, on a vocal-free version of our dataset. The results provided in Table A.3 show a worse FAD and KL on our in domain test set. On MusicCaps, the FAD is improved by using DAC, although the KL is worse, as well as the subjective evaluations. The EnCodec model used in this work was specifically

---

[12]Using the public implementation github.com/descriptinc/descript-audio-codec.

Table A.3: We test replacing EnCodec with DAC [Kumar et al., 2023] using their implementation. DAC is a 44.1 kHz model with 9 codebooks and a frame rate of 86 Hz. Those models are trained on a vocal-free version of our dataset, hence the objective metrics will not match those reported in the other tables. We report objective metrics both on our in domain test set and MusicCaps [Agostinelli et al., 2023], and subjective metrics only on MusicCaps.

| | In Domain Test Set | | MUSICCAPS Test Set | | |
|---|---|---|---|---|---|
| MODEL | $\text{FAD}_{\text{vgg}} \downarrow$ | $\text{KL} \downarrow$ | $\text{FAD}_{\text{vgg}} \downarrow$ | $\text{KL} \downarrow$ | $\text{OVL.} \uparrow$ |
| MUSICGEN + DAC small | 3.45 | 0.58 | 4.46 | 1.35 | $83.32_{\pm 0.95}$ |
| MUSICGEN + DAC medium | 2.42 | 0.57 | **4.32** | 1.30 | $84.46_{\pm 0.97}$ |
| MUSICGEN + EnCodec small | 0.67 | 0.54 | 5.26 | 1.27 | $84.69_{\pm 0.90}$ |
| MUSICGEN + EnCodec medium | **0.49** | **0.52** | 5.05 | **1.23** | $\mathbf{86.09}_{\pm 0.88}$ |

designed to operate at a lower frame rate (50 Hz) than DAC (86 Hz), thus reducing by 40% the inference runtime for producing the same audio. Note finally that DAC was trained on a different dataset than EnCodec, and further experiments would be required to understand exactly what influences the fitness of such compression models to be used with auto-regressive language models.

**Effect of D-Adaptation.**

D-Adaptation is a novel automated way of selecting the overall learning rate of the Adam optimizer, i.e. its $\alpha$ parameter, dynamically throughout the training, introduced by Defazio and Mishchenko [2023]. We observed improved convergence for the 300M parameter model, although for larger models, e.g. 1.5B and 3.3B, we observed the automated rule led to deteriorated performance, both on the train and validation set. Further investigation is required to better understand the effect of D-Adaptation, and whether it can scale to the largest model. The convergence for both methods can be observed on the train and validation set in Figure A.4.

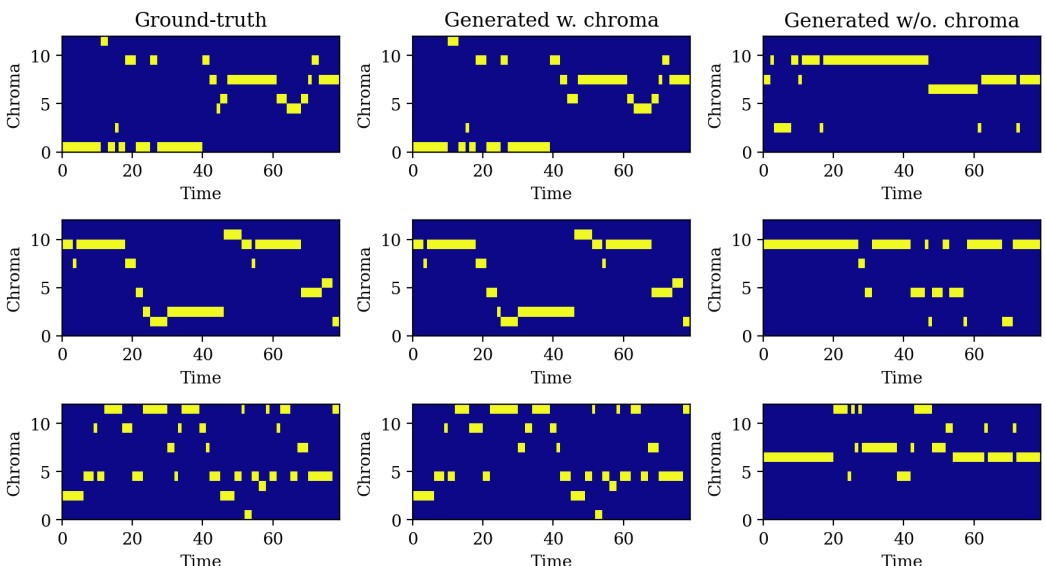

Figure A.2: Visualization of quantized chromagram bins over time from reference melody (left), generated music conditioned on chroma and text (middle) and generated music with text-only conditioning (right). Each row is generated using a different chroma condition, all rows share the same text condition: "90s rock song with electric guitar and heavy drums". We observe strong adherence to the input melody for the music samples generated with chroma conditioning while rendering novel styles guided by the input text.

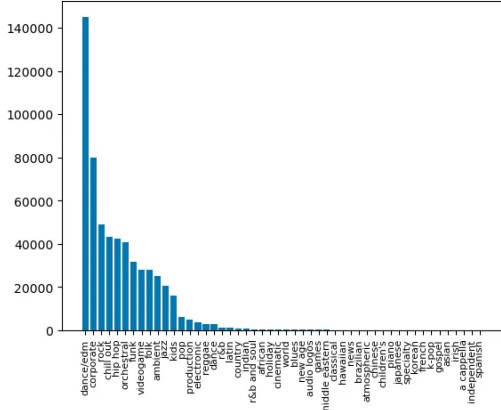

Figure A.3: Histogram of the top 50 musical genres in the training dataset.

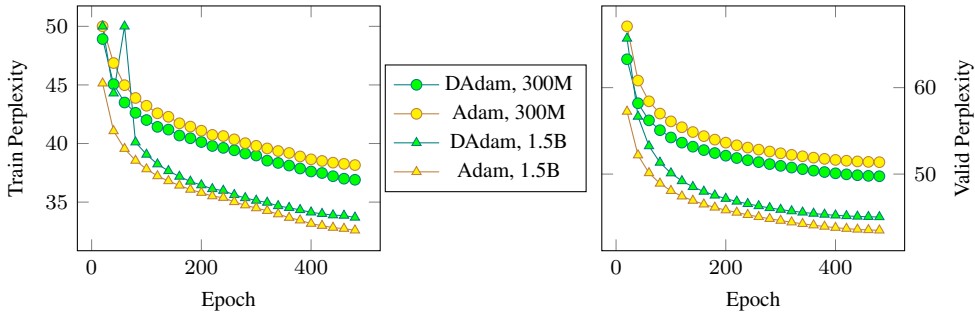

Figure A.4: Comparison of Adam and Adam with D-Adaptation [Defazio and Mishchenko, 2023]. While D-Adaptation provided consistent gains for the 300M parameters model, we observed worse convergence both on the train (left) and validation (right) set for the 1.5B parameters model.

