# OpenReview forum: "Simple and Controllable Music Generation"
_NeurIPS.cc/2023/Conference — NeurIPS 2023 poster_

### Official Review · Reviewer_gnNB · 2023-06-22

**Soundness:** 3 good
**Presentation:** 4 excellent
**Contribution:** 3 good
**Rating:** 8
**Confidence:** 4

**Summary:**

This paper introduces MusicGen, an innovative single-stage transformer language model for conditional music generation, using either text or melody as a conditioning factor. The authors focus on exploring different approaches of codebook interleaving patterns and propose an efficient approach of Delay Pattern. Experiments were performed with state-of-the-art conditional music generation models, and multiple ablation studies aids the efficacy of the proposed approach. The authors have further contributed to the community by releasing demo samples and inference code to the public.

**Strengths:**

- The paper is presented clearly, which greatly aids comprehension of the presented methods and results.
- The proposed *Delay Pattern* is a novel approach that shows strengths in computational time. It’s surprising to see the decoder interpret the residual codebooks of the Delay Pattern, even when the delayed codebooks are summed and used as input. A more in-depth theoretical explanation of this would be valuable.
- The empirical results are impressive, where the evaluation against state-of-the-art text to music generation systems provides solid evidence for the model's performance.
- The authors' decision to open-source the code and model weights could significantly impact both academia and industry. I’m already seeing some online applications utilizing MusicGen's API.

**Weaknesses:**

- The reproduced *Valle-E pattern* among the *codebook interleaving patterns* seems inconsistent with the original manuscript [A].
In the Section **Introduction**, the authors correctly mention the codebook generation steps of *Valle-E* as “(i) modeling the first stream of tokens only; (ii) then, apply a post-network to jointly model the rest of the streams in a non-autoregressive manner.”
Yet, the authors reproduced the *Valle-E* pattern as the step (ii) being decoded in an autoregressive manner. The correct pattern of *Valle-E* would be to use two different decoders and perform step (ii) in a nonautoregressive manner (given the autoregressively generated tokens from step (i)), resulting in sequence steps of $n+3$.
    - Thus, this discrepancy should be addressed either by 1. correcting the *Valle-E* pattern methodology or 2. simply renaming the “Valle-E Pattern” in the manuscript. #1 option would be to rectify *Figure 1, Section 2.2, Equation (6), Table 3 (Nb. steps)*, and updating the results of *Table 3*.

- Some aspects of the paper lack details and justifications of method choice.
    1. According to the ablation study results of *text augmentation* Table A.2, the best performing method is applying only the *condition merging*. Since the authors are using *Condition merging + Text-norm. + Word dropout* as the final text augmentation strategy, they should provide a justification for this.
    2. The final choice of text encoder is unclear. Although the authors have stated that they performed an ablation study using all three different text encoders (T5, Flan-T5, and CLAP), the reviewer couldn’t find an indication which model was selected as the final text encoder. Furthermore, the results from the ablation study Table A.1 does not align with any of the scores from Table 1 & 2. If the authors used different configurations for Table A.1, they should mention it in the manuscript.

- typo at line 196: “hope size” → “hop size”

[A] Wang, Chengyi, et al. "Neural Codec Language Models are Zero-Shot Text to Speech Synthesizers." *arXiv preprint arXiv:2301.02111* (2023).

**Questions:**

Please see the concerns raised at **Weakness** section. + theoretical explanation of the decoder interpreting *Delay Pattern* mentioned at **Strengths** section.

**Limitations:**

The paper comprehensively outlines its limitations and discussions in the manuscript, which align with the reviewer's.

---

> ### Author Rebuttal · Authors · 2023-08-09
>
> **_Regarding the VALL-E codebook pattern_:** Indeed, we incorrectly describe the “VALL-E” codebook patterns as corresponding to VALL-E. We renamed it to “flat 1st codebook then parallel”.
>
> **_Regarding text augmentations_:** During model development, we experimented with condition merging, word dropout, and text normalization. For the final models, we use condition merging + word dropout as it gives the best overall performance. We agree the description in the text is confusing and we will clarify that for the final manuscript. To better support our design choice, we provide here a human study (OVL. and REL.) in the table below.
>
> ### Text augmentations Human study
>
> | Configuration                                 | FAD      | KL   | CLAP sc. | OVL.           | REL.           |
> |-----------------------------------------------|----------|------|----------|----------------|----------------|
> | No augmentation                               | 3.68     | 1.28 | **0.31** | **83.4** ± 1.4 | 81.2 ± 1.3     |
> | Condition merging                             | **3.28** | 1.26 | **0.31** | 82.6 ± 1.4     | 84.5 ± 1.2     |
> | Condition merging + Text-norm.                | 3.78     | 1.30 | 0.29     | 80.6 ± 2.1     | 82.4 ± 1.1     |
> | Condition merging + word dropout              | 3.31     | 1.31 | 0.30     | 82.5 ± 1.6     | **85.3** ± 1.0 |
> | Condition merging + Text-norm. + Word dropout | 3.41     | 1.39 | 0.30     | 81.2 ± 1.9     | 84.3 ± 1.6     |
>
> Results suggest condition merging + word dropout provides comparable generation quality but better text relevance scores.
>
> **_Regarding the used text encoder_:** We use the T5 text encoder as noted in line 192. Considering all metrics, the T5 model gets the best overall performance (Table A.1). The configuration used in Table A.1 is indeed different. For Table A.1, as this is an ablation study, we use a 300M parameters model (see lines 170-171). We now indicate clearly when an Ablation is done with a 300M parameters model, and in Table A.1 we further repeat that T5 is the encoder used for the main models in the caption of Table A.1.
>
> **_Regarding typos_:** Thanks! We will fix the typo.

---

### Official Review · Reviewer_7q8g · 2023-06-27

**Soundness:** 4 excellent
**Presentation:** 3 good
**Contribution:** 3 good
**Rating:** 7
**Confidence:** 3

**Summary:**

MusicGen is an auto-regressive architecture for music audio generation conditioned on textual descriptions and an optional melody. The key proposal is a generic formulation of audio codebook interleaving strategy, which enables parallel code streams to be processed with a simple single-stage Transformer decoder while balancing efficiency with inter-stream dependency. Through extensive evaluation, MusicGen demonstrates a superior performance in acoustic coherency and semantic faithfulness compared to previous models. Sufficient ablation studies also validate its design choices for text pre-processing, melody conditioning, code interleaving pattern, and model scale.

**Strengths:**

* **Unsupervised melody conditioning**: The paper uses quantized chromagrams to capture the most salient melodic features to conduct melody conditioning. To suppress interference with low-frequency instruments, it further leverages a pretrained source separation model to detach those bands before extracting chromagram. Such an unsupervised scheme is efficient, sound, and working reasonably well.

* **Ablation study**: Extensive empirical studies are conducted to validate the key design choices, which are beneficial for the reference of follow-up research.

* **Compelling performance**: Demos demonstrate a compelling acoustic and semantic quality. It is good to know that code and models will be open, for which a good impact of this work can be expected.


**Weaknesses:**

* **Long-term music structure**: The paper demonstrates several long generation examples created using windowed sampling, showcasing the ability of MusicGen to produce remarkably authentic grooves and looping patterns. However, one noticeable weakness lies in the relative absence of a distinct phrasal or sectional structure in the long run, which may lead to a lack of sense of musical development. To tackle such problem may require more musically insightful inductive bias in the framework design.

* **Fine-grained control**: The text-conditional architecture of MusicGen focuses on global semantic control, which essentially guarantees a globally consistent “picture” throughout the generated piece. On the other hand, musicians sometimes intentionally want to create variations and inconsistency to make the piece more impressive and contagious. Achieving such purpose would require a more fine-grained controlling scheme, which is not currently supported by MusicGen.


**Questions:**

* There seems a notation clash on the variable $N$. On line 74, $N$ is defined as codebook size, but is later used to represent code sequence length in line 98, line 145, and the caption of Figure 1.

* Line 95: Should the autoregressive steps of MusicLM be $df_r\cdot K$, rather than $df_s\cdot K$?

* Line 177: What is the purpose of applying EMA to the Transformer model? Shouldn’t it be part of the audio tokenization model for updating the codebook?


**Limitations:**

In my view, the main potential limitation of this work lies in long-term music structure and fine-grained control, which has been detailed in the Weaknesses section. The latter has also been discussed in the paper. However, I understand that this paper focuses on text semantic control (with already compelling results) and those areas are currently out of the scope.

---

> ### Author Rebuttal · Authors · 2023-08-09
>
> **_Regarding long-range musical structure_:** We agree that evaluating musical structure in the generated music is interesting and important for music generation. However, it is far from trivial as the generated output is audio samples and not interpretable discrete representations (like midi). Developing such metrics is an important future research direction, we hope the community (including us) will pursue it. Nevertheless, we believe MusicGen is an important step in the right direction of developing high-quality music generation models.
>
> **_Regarding fine-grained control_:** We agree fine-grained control can benefit the music industry and specifically music creators. However, we believe global control is also important and can benefit creators who are not expert musicians. Nevertheless, we believe MusicGen is an important step in the right direction toward developing controllable music generation.
>
> **_Regarding notations_:** We fixed this notation. We renamed the $N$ which corresponds to the time steps into $T$ in line 98 and Figure 1.
>
> **_Regarding line 95, MusicLM_**: Thanks for the correction! We fixed it for the final manuscript.
>
> **_Regarding EMA_:** During preliminary experiments, we notice using an exponential moving average during model training smoothes out the validation loss. Such a technique was also used in prior work with transformers, see [1].
>
> [1] Touvron, Hugo, et al. "Training data-efficient image transformers & distillation through attention." International conference on machine learning. PMLR, 2021.

---

> > ### Comment · Reviewer_7q8g · 2023-08-17
> >
> > Thanks for your response to my review.
> >
> > At a system level, the technical contribution of the paper is undoubtedly solid. While the work may exhibit a certain lack of scientific novelty and deeper insights into the music generation task, I am convinced that "MusicGen is an important step in the right direction". The additional experimental results on stereo generation and memorization analysis would undoubtedly deepen the potential impact of this work. I appreciate the efforts of the authors and would maintain my recommendation of acceptance.

---

### Official Review · Reviewer_VPXA · 2023-07-04

**Soundness:** 3 good
**Presentation:** 4 excellent
**Contribution:** 3 good
**Rating:** 5
**Confidence:** 4

**Summary:**

The paper proposes a single-stage music generation model that can input text or melody. The proposed approach uses tokens from pre-trained neural audio codec tokens with multiple residual vector quantizers and investigates efficient language modeling to reduce the length of autoregressive steps. Experiments on large-scale datasets show better generative results than some baselines.

**Strengths:**

1. It is challenging to achieve strong performance and efficiency simultaneously, but this paper shows these could be achieved elegantly.

2. The samples in supplementary have high audio quality and text-music coherence. I could perceive the difference in comparison to other approaches.

3. The paper is generally written well. The motivation and demonstrations are clear.

4. I am glad that the code and models will be available for future comparison and reproduction.

**Weaknesses:**

1. While the generative samples are impressive, I could see the technical novelty in this paper is limited. The audio tokens or EnCodec models are widely used in various applications. The 'parallel' or 'delay' codebook interleaving patterns are proposed in previous papers. The text2music or melody2music tasks have been established. I was having a difficult time capturing the core contribution of this paper. From my perspective, the main contribution of this paper would be to perform extensive experiments, analyze the results and make comparisons to demonstrate the critical components for efficient modeling. From this aspect, additional experiments (discussed in the questions) might be needed to convince me fully.

2. Nowadays, we know data is a core part of reaching successful training, and obviously, the collected licensed music data probably won't be released. It is not a unique situation, but reproduction and future comparison will become an issue, and I wish authors could discuss this aspect. Besides, a genre distribution of the collected data might be helpful.

3. Evaluation of generated music is another tricky thing. I understand that FAD, KL-div, CLAP consistency, and subjective evaluations are almost the standard evaluation metrics, but none consider whether the musical structure makes sense. I am wondering when the evaluation gap between symbolic (midi) based and audio-based methods could be consistent.

4. In the MusicLM paper, there is a specific section about data memorization, which could be problematic, especially for licensed data. I want authors to discuss how such an issue could be avoided/improved as the technical components become mature.

**Questions:**

1. How does the number of codebooks affect the modeling? The original Encodec has 8 codebooks, and SoundStream has 12 codebooks. The 'flatten' idea becomes much more computationally expensive, but how could you ensure it would degrade the performance when the number of codebooks increases? Similarly, the codebook size also could change the difficulty of the optimization.

2. the authors tried various text encoders in the main paper, but the generated samples are not described. In particular, I am interested in comparing CLAP text embedding and T5.

**Limitations:**

As I mentioned in the weakness, data memorization could be an issue and I am looking forward to the author's response.

---

> ### Author Rebuttal · Authors · 2023-08-09
>
> **_Regarding the contribution of the proposed method_:** The novelty and contribution of our work are designing a simple and efficient auto-regressive model to perform text-to-music generation. Unlike prior work, which consists of a cascade of models using either super-resolution or semantic tokens, we present a single-stage language model through an efficient codebook interleaving strategy. With the codebook-interleaving patterns, we provide a framework that generalizes prior work in the field. Additionally, we present a single model to perform both text and melody-conditioned generation and demonstrate that the generated audio is coherent with the provided melody and faithful to the text-conditioning information.
>
> Following the reviewer's concern, we additionally include an extension of MusicGen to stereophonic music generation, making it the first model to generate stereophonic music conditioned on textual descriptions. OVL. and REL. scores can be seen in the table above (in response to reviewer A118) (and inside the attached rebuttal PDF file, Table 1), samples were shared with the AC through an external link as per the NeurIPS guidelines. Due to all of the above, we believe our work is novel and the contributions are important and would be valuable to the community.
>
> **_Regarding the used dataset_:** our dataset consists of 20K hours of licensed music, comprising 10k high-quality music data, ShutterStock, and Pond5 music data collections, which are available for licensing (lines 202-206). As per the reviewer's request, we provide a genre distribution of the training set in the attached rebuttal PDF file (Figure 3). Regarding comparison to MusicGen, we will open-source both training code and pre-trained models, so future research could: (1) directly compare the MusicGen models, and (ii) build new and improved music generation models on top of MusicGen.
>
> **_Regarding evaluation function_:** We agree that evaluating musical structure in the generated music is interesting and important for music generation. However, this is far from trivial as the generated output is audio samples and not interpretable discrete representations (like midi). Developing such metrics is an important future research direction, we hope the community (including us) will pursue it. Nevertheless, we believe MusicGen is an important step in the right direction of developing high-quality and controllable music generation models.
>
> **_Regarding memorization metrics_:** As per the reviewer’s request, we follow Agostinelli et al. [2023], and analyze the training data memorization abilities of MusicGen. We consider the first stream of codebooks from MusicGen as it contains the coarser grain and most important regarding the generated audio. We randomly select $N=20,000$ examples from our training set and for each one, we feed the model with a prompt of EnCodec codebooks corresponding to the original audio and the conditioning information. We generate a continuation of 250 audio tokens (corresponding to 5 seconds) using greedy decoding. We report exact matches as the fraction of examples for which the generated audio tokens and source audio tokens are identical over the whole sequence. In addition, we report partial matches with the fraction of examples for which the generated and source sequences have at least $80\%$ of the audio tokens matching. The Figure can be seen in the attached rebuttal PDF file (Figure 2). As can be seen, the exact matches and partial matches are almost zero under all evaluated settings.
>
> **_Regarding the number of codebooks (question 1)_:** During early experiments, we compared several numbers of codebooks and codebook sizes. We noticed that larger codebooks (e.g. 2048) would improve quality but not increase it beyond that point (we tested 4096 and 8192). Early experiments indicated that using more codebooks (e.g. 8) was detrimental to MusicGen models with 1.5B parameters. However, as depicted in Table 1 in the Rebuttal Document, we show that we can extend to 8 codebooks to stereo data, using 4 codebooks for the left channel, and 4 codebooks for the right channel. In that case, we noticed no regression compared with the 4 codebooks model (e.g. downmixing the output to mono to compare to the previously trained mono models), while the model was clearly able to model the stereo image of the generated music extracted, as measured by subjective evaluations.
>
> **_Regarding samples for different text encoders (question 2)_:** As per the reviewer’s request, we shared samples for various text encoders. Specifically, we share samples for T5, Flan-T5, and CLAP using 12 codebooks (previously just CLAP, as done in MusicLM), CLAP using 24 codebooks, and CLAP without quantization. Interestingly, we observe that the quantization of the CLAP embedding as done in MusicLM is detrimental to the final score of the model. Samples were shared with the AC through an external link as per the NeurIPS guidelines. We additionally provide results for the different setups in the table below:
>
> | Text Encoder                | FAD      | KL       | Clap sc. | OVL.           | REL.           |
> |-----------------------------|----------|----------|----------|----------------|----------------|
> | T5 (default)                | **3.12** | **1.29** | 0.31     | 84.9 ± 1.8     | **82.5** ± 1.3 |
> | FLAN-T5                     | 3.36     | 1.30     | 0.32     | **86.3** ± 1.8 | 80.8 ± 1.9     |
> | CLAP (RVQ n_q=12 bins=1024) | 4.23     | 1.53     | 0.32     | 79.8 ± 1.8     | 77.3 ± 1.5     |
> | CLAP (RVQ n_q=24 bins=1024) | 4.18     | 1.47     | 0.32     | 82.0 ± 1.4     | 76.7 ± 1.3     |
> | CLAP (no quantizer)         | 5.13     | 1.53     | **0.34** | 84.5 ± 1.2     | 80.2 ± 1.      |

---

> > ### Comment · Reviewer_VPXA · 2023-08-18
> > **Post Rebuttal**
> >
> > Thank the authors for the responses. All of my questions are answered with details and well addressed. Now I agree that the paper is technically solid, and the contribution is clear. I will recommend accepting the paper.

---

### Official Review · Reviewer_A118 · 2023-07-06

**Soundness:** 3 good
**Presentation:** 2 fair
**Contribution:** 2 fair
**Rating:** 5
**Confidence:** 4

**Summary:**

This paper introduces MUSICGEN, an approach for generating music conditioned on either text or melody representation. MUSICGEN consists of a single-stage transformer language model (LM) augmented with efficient token interleaving patterns, eliminating the requirement of employing multiple cascaded models. The authors extensively evaluate the proposed approach through a combination of automatic evaluations and human studies, demonstrating its superiority over the evaluated baselines on a widely-used text-to-music benchmark.

**Strengths:**

1. The experimental setup and results presented in the paper are robust and persuasive, providing strong evidence for the proposed approach.
2. The paper includes a detailed and comprehensive discussion of the patterns related to the multi-codebook technique, enriching the understanding of this aspect of the research.
3. The demos in supplementary materials are impressive.

**Weaknesses:**

1. The novelty of the proposed approach is somewhat limited, and the motivation for the research is not clearly articulated. Additionally, certain design choices in the methodology lack explanation, necessitating further elaboration and clarification.
2. Some notations in the formulas are confusing, such as the usage of $\widetilde{U}$ in Equation (2) and $\hat{p}$ in Line 85. These notations should be introduced and defined more explicitly to avoid confusion.
3. Attention to writing details is required. For instance, when referencing equations in the main text, only the number is provided without explicitly mentioning “Equation” (e.g., Line 89, Line 101).

**Questions:**

Despite having some concerns regarding the motivation and novelty, I am impressed by the thoroughness of the experiments. If the authors can further improve the writing quality, I am willing to give a higher score.

---

> ### Author Rebuttal · Authors · 2023-08-09
>
> **_Regarding novelty and motivation:_** The novelty and motivation of the proposed method (MusicGen) are designing a simple and efficient auto-regressive model to perform text-to-music generation. Unlike prior work, which consists of a cascade of models using either super-resolution or semantic tokens, we present a single-stage language model through an efficient codebook interleaving strategy. Additionally, we present a single model to perform both text and melody-conditioned generation and demonstrate that the generated audio is coherent with the provided melody and faithful to the text-conditioning information.
>
> Following the reviewer's concern, we additionally include an extension of MusicGen to stereophonic music generation, making it the first model to generate stereophonic music conditioned on textual descriptions. We encode separately the left and right channels using the same EnCodec model and use a specifically designed codebook interleaving pattern depicted in Figure 1 of the rebuttal document. OVL. and REL. scores can be seen below (and inside the attached PDF file), samples were shared with the AC through an external link as per the NeurIPS guidelines. Due to all of the above, we believe our work is novel and the contributions are important and would be valuable to the community.
>
>
> ### Stereo Model Human Study.
>
>
> | Cb. Pattern          | Stereo? | OVL.           | REL.           |
> |----------------------|---------|----------------|----------------|
> | mono delay           | ✗       | 85.0 ± 1.6     | **80.6** ± 1.2 |
> | stereo partial delay | ✗*      | 84.5 ± 1.8     | 79.4 ± 1.2     |
> | stereo partial delay | ✓       | **86.7** ± 1.1 | **80.4** ± 1.1 |
> | stereo delay         | ✓       | 85.5 ± 1.2     | 78.3 ± 1.2     |
>
>
> *: downmixed to mono
>
> **_Regarding notations and writing quality_:** We thank the reviewer for their feedback and suggestions. We will fix all writing issues raised by the reviewer, including Equation references and method notations.
> When introducing $\tilde{U}$, we clarified as “Let us build a second sequence of random variables $\tilde{U}$ using the auto-regressive density $p$, e.g. we define recursively $\tilde{U}_0 = 0$ and for all t > 0, …
> We introduce $\hat{p}$ as an estimate of $p$ using a deep learning model. We clarified “This means that if we can fit a perfect estimate $\hat{p}$ of $p$ with a deep learning model, then we can fit exactly the distribution of $U$.”

---

### Author Rebuttal · Authors · 2023-08-09

We would like to thank the reviewers for their detailed reviews and valuable feedback. We are happy the reviewers found our experimental setup and results robust and persuasive. We are also glad the reviewers found our method to have both strong performance and efficient modeling. We address each of the reviewers' questions and concerns in a separate comment below.

---

### Decision · Program_Chairs · 2023-09-21

**Decision:**

Accept (poster)

**Comment:**

This paper proposes MusicGen, a single-stage Transformer language model to perform text-to-music generation, with efficient token interleaving patterns. All reviewers are positive about this paper. Although some reviewers have challenges with the technical novelty and motivations, the authors provide a good explanation, and let the reviewer support that this work is ''an important step in the right direction''.
The open-source code and checkpoint also resolve the concerns about the unavailability of the training data to some extent. Overall, it is a good paper and I recommend an acceptance.